# Electron Spin Correlations: Probabilistic Description and Geometric Representation

**DOI:** 10.3390/e24101439

**Published:** 2022-10-09

**Authors:** Ana María Cetto

**Affiliations:** Instituto de Física, Universidad Nacional Autónoma de México, Mexico City 04510, Mexico; ana@fisica.unam.mx

**Keywords:** spin correlation, conditional probability, contextuality, physical picture

## Abstract

The electron spin correlation is shown to be expressible in terms of a bona fide probability distribution function with an associated geometric representation. With this aim, an analysis is presented of the probabilistic features of the spin correlation within the quantum formalism, which helps clarify the concepts of contextuality and measurement dependence. The dependence of the spin correlation on *conditional* probabilities allows for a clear separation between system state and measurement context; the latter determines how the probability space should be partitioned in calculating the correlation. A probability distribution function ρ(ϕ) is then proposed, which reproduces the quantum correlation for a pair of single-particle spin projections and is amenable to a simple geometric representation that gives meaning to the variable ϕ. The same procedure is shown to be applicable to the bipartite system in the singlet spin state. This endows the spin correlation with a clear probabilistic meaning and leaves the door open for a possible physical picture of the electron spin, as discussed at the end of the paper.

## 1. Introduction

The extent to which quantum mechanics require a different kind of probabilities from those used in classical statistical mechanics is still an open question. Clarification of the issue is not only of fundamental importance for a better understanding of quantum theory and a demystification of the quantum phenomenon, including issues such as nonlocality, acausality or the absence of realism; it also has important implications for the development and extension of probability theory with a view to its applications in other areas, as complex and diverse as epidemiology, finances, game theory and cognitive science (see, e.g., [1,2] and references therein).

The present paper addresses the question for the specific case of the electron spin correlation in an effort to clarify whether the unusualness of the quantum formalism is rooted in its probabilistic framework and, most importantly, whether it implies the need to renounce basic principles that hold for the rest of physics. For this purpose, a detailed analysis is presented for the probabilistic features of the spin correlation contained in the quantum formalism. Their concrete realization in the form of a bona fide probability distribution is proposed. Further, this probability distribution is shown to be amenable to a spatial representation, thus paving the way for a possible physical image of the electron spin.

Two conceptual elements that are shown to play a central role in the analysis are the context and the conditional probabilities. A distinction is made between the notion of context used here to refer to the measurement that is carried out—i.e., *what* is being measured—as opposed to the notion of contextuality frequently used in quantum measurement theory to refer to the result of a measurement being dependent on which other quantity has been measured. By the same token, conditional probabilities as discussed here are probabilities conditioned by the context. Such context-conditioning is connected with the specific partitioning of the probability space, as has been shown in previous work [3].

Consideration of the context dependence of the probability space partitioning is essential to arrive at a geometric representation of the proposed probability distribution function ρ(ϕ), whose argument ϕ varies at random within its integration range. This hidden-variable description is shown to reproduce the probabilistic features [4] and the quantum result for the one-electron spin correlation. An analagous procedure is shown to be applicable to the bipartite singlet spin case. That both cases can be dealt with following a similar approach is due to the use of conditional probabilities in calculating the respective correlations. Further to endowing the probabilities with a concrete meaning, the results obtained open the possibility of an understanding of the physics that underlies the quantum description. A proposal in this regard is advanced at the end of the paper in light of recent experimental evidence pointing to a finer dynamics of the spinning electron, which requires further investigation.

The paper is structured as follows. Section 2 starts with the introduction of an algebraic representation of the spin-projection probabilities for the one-electron spin case, which serves to discuss the notions of contextuality and conditioned probabilities. This representation is shown briefly to reproduce the basic probabilistic properties predicted by the quantum formalism for the electron spin correlation. A central feature of the algebraic approach is the clear separation of the context (what is being measured) from the state of the system (in which it is measured). The quantum description of the spin correlation is shown to imply a context-dependent disaggregation of the probability space into mutually exclusive subspaces. In Section 3, a probability distribution function ρ(ϕ) is introduced that reproduces the quantum probabilistic results. This distribution function is shown in Section 4 to be amenable to a geometric representation that gives meaning to the random variable ϕ. In Section 5, the same probabilistic approach is shown to be applicable to the bipartite singlet spin state and to correctly reproduce the quantum correlation. The paper concludes with a discussion on the possibility of a physical picture for the electron spin.

## 2. The Spin-1/2 Particle

### 2.1. Analysis of Contextuality

In a recent article [5], Grangier introduced a “principle of contextual quantization”, meaning that *whatever the context, a measurement on a given system gives one modality among N possible ones, where the value of N characterizes the system. These N modalities are mutually exclusive, i.e., only one can be realized at a time*.

Thus, for example, the projection of an electron spin along an arbitrary direction a gives either +1 or −1. Since ± are the only possible outcomes, N=2. Assume first that the result of the projection along a is +1; if the spin is measured again along a, the result +1 is again obtained. If, however, the projection is measured along a different direction b, one gets −1 with a certain probability. This can be expressed by means of a 2×2 matrix of probabilities that depends on the two directions a and b, the rows referring to the possible signs of a and the columns to those of b:(1)$$P\left(b,a\right)=\left(\begin{array}{cc} P_{ab}\left(+|+\right) & P_{ab}\left(-|+\right)\\ P_{ab}\left(+|-\right) & P_{ab}\left(-|-\right) \end{array}\right),$$
with $P_{ab}\left(b|a\right)$ being the probability of *b* conditioned by the value of *a*. Thus for instance, $P_{ab}\left(-|+\right)$ is the probability that, given a +1 projection along a, the projection along b is −1. Clearly, since the projection along b must be either +1 or −1,
(2a)$$P_{ab}\left(+|+\right)+P_{ab}\left(-|+\right)=1,$$
and
(2b)$$P_{ab}\left(+|-\right)+P_{ab}\left(-|-\right)=1.$$

These probabilities are invariant under an inversion of the sense of the directions a and b that interchanges all the plus and minus signs
(3a)$$P_{ab}\left(+|+\right)=P_{ab}\left(-|-\right),$$
and
(3b)$$P_{ab}\left(+|-\right)=P_{ab}\left(-|+\right).$$

The matrix P(a,b) is therefore symmetric, i.e., P(a,b)=P(b,a), whence a and b may be freely interchanged. Moreover, it is doubly stochastic [2], because both the rows and the columns add to 1.

Notice that the matrix coefficients represent *conditional* probabilities, with the upper ones referring to the (+ or −) projections along b conditioned by the +1 projection along a and the lower ones by the −1 projection along a. The corresponding *joint* probabilities are given by expressions of the form [6]
(4)$$P_{ab}\left(++\right)=P_{a}\left(+\right)P_{ab}\left(+|+\right),\;P_{ab}\left(-+\right)=P_{a}\left(+\right)P_{ab}\left(-|+\right),$$
where $P_{a}\left(+\right)$ is the (total) probability of the projection along a being +1, and similarly for the lower pair. Thus, the conditional probabilities $P_{ab}\left(+|+\right)$, $P_{ab}\left(-|+\right)$ restrict the sample space to the situation in which the projection along a is +1, and similarly for $P_{ab}\left(-|-\right)$, $P_{ab}\left(+|-\right)$. This will be important for the discussion in Section 4. The total probability is the sum of the respective joint probabilities; thus, for instance,
(5)$$P_{b}\left(+\right)=P_{a}\left(+\right)P_{ab}\left(+|+\right)+P_{a}\left(-\right)P_{ab}\left(+|-\right).$$

Clearly,
(6)$$P_{a}\left(+\right)+P_{a}\left(-\right)=1,\;P_{b}\left(+\right)+P_{b}\left(-\right)=1.$$

The correlation of the projections is given by the formula
(7)$$C\left(a,b\right)=\frac{P_{ab}\left(++\right)+P_{ab}\left(--\right)-P_{ab}\left(-+\right)-P_{ab}\left(+-\right)}{P_{ab}\left(++\right)+P_{ab}\left(--\right)+P_{ab}\left(-+\right)+P_{ab}\left(+-\right)}.$$

On account of Equations (2)–(Equation 6), the sum of the joint probabilities in the denominator gives 1, and Equation (Equation 7) simplifies into
(8)$$C\left(a,b\right)=P_{ab}\left(+|+\right)-P_{ab}\left(-|+\right)=P_{ab}\left(-|-\right)-P_{ab}\left(+|-\right).$$

Notice that, by involving the conditional probabilities only, this result is independent of the total probabilities $P_{a}\left(\pm \right)$, $P_{b}\left(\pm \right)$. This is an important feature of the matrix of probabilities, as it means that it applies to the spin projections along a and b as described above, *regardless of the spin state*. In practical terms, the terms contained in Equation (Equation 8) depend on the arrangement of the measuring devices, not on the preparation of the spin to be measured. Briefly stated, P(a,b), and hence also C(a,b), refer to the *contextuality* of the measurements, in line with the meaning of the term ’context’ used in Refs. [2,5].

### 2.2. Spin-1/2 Projection Probabilities

To calculate the conditional probabilities for the one-electron spin case, we use the standard expressions for the bases of spin state vectors along two arbitrary directions a and b lying on the same vertical plane and forming angles $\theta_{a}$ and $\theta_{b}$, respectively, with the *z* axis. In terms of $\vartheta_{a,b}\equiv \theta_{a,b}/2$,
(9)$${|+\rangle }_{a}=\left(\begin{array}{c} \cos\vartheta_{a}\\ -\sin\vartheta_{a} \end{array}\right),\;{|-\rangle }_{a}=\left(\begin{array}{c} \sin\vartheta_{a}\\ \cos\vartheta_{a} \end{array}\right),$$
and similarly for ${|\pm \rangle }_{b}$. This gives, with $\vartheta_{ba}=\vartheta_{b}-\vartheta_{a},$
(10a)$${}_{b}\langle +{|+\rangle }_{a}={}_{b}\langle -{|-\rangle }_{a}=\cos\vartheta_{ba},$$
(10b)$${}_{b}\langle +{|-\rangle }_{a}=-{}_{b}\langle -{|+\rangle }_{a}=-\sin\vartheta_{ba}.$$

The conditional probabilities are therefore given by
(11a)$$P_{ab}\left(+|+\right)=P_{ab}\left(-|-\right)=\cos^{2}\vartheta_{ba},$$
(11b)$$P_{ab}\left(+|-\right)=P_{ab}\left(-|+\right)=\sin^{2}\vartheta_{ba},$$
whence Equation (Equation 1) becomes
(12)$$P\left(b,a\right)=\left(\begin{array}{cc} \cos^{2}\vartheta_{ba} & \sin^{2}\vartheta_{ba}\\ \sin^{2}\vartheta_{ba} & \cos^{2}\vartheta_{ba} \end{array}\right).$$

From Equation (Equation 8), we obtain for the correlation of the spin projections
(13)$$C_{Q}\left(a,b\right)=\langle \psi |\left(\hat{\sigma }\cdot b\right)\left(\hat{\sigma }\cdot a\right)|\psi \rangle$$
the well-known result for the quantum correlation,
(14)$$C_{Q}\left(a,b\right)=\cos^{2}\vartheta_{ba}-\sin^{2}\vartheta_{ba}=\cos\theta_{ba},$$
regardless of the spin state $|\psi \rangle$.

### 2.3. On the ‘Quantumness’ of Spin Probabilities

The mathematical element represented by Equation (Equation 1), with its associated properties discussed above, is, according to Grangier [5], a ‘fundamentally quantum idea’, because with a couple of simple consistency arguments it leads to the inevitable conclusion that the only possible theory is quantum mechanics.

The first consistency argument refers to the sum of the projectors, which must be equal to 1, as indicated in Equations (2), for any measurement context. The appeal made in [5] to Gleason’s theorem does not apply to the present case, in which we are dealing with a two-dimensional Hilbert space [7,8]. It would seem, therefore, that we need to resort to the Kochen–Specker theorem [9], which excludes any non-contextual hidden-variable theory able to reproduce the quantum results, thus assigning a seal of uniqueness to quantum probabilities. This stresses the relevance of establishing a clear definition of what is meant by contextual, a point to which we will return in the following sections.

The second consistency argument in [5] refers to the unitarity of the transformations between projectors, which is necessary to preserve the mutually exclusive character of events in each context [10]. That this condition is satisfied can be proved by the mathematical procedure of associating with the probability matrix P(b,a), given by (Equation 12), an orthogonal matrix $F_{ba}$ whose elements are the square roots of the coefficients of P(b,a),
(15)$$F_{ba}=\left(\begin{array}{cc} \cos\vartheta_{ba} & -\sin\vartheta_{ba}\\ -\sin\vartheta_{ba} & -\cos\vartheta_{ba} \end{array}\right).$$

Indeed, a change of measuring context, from (a,b) to (a,c), with $\vartheta_{ca}=\vartheta_{c}-\vartheta_{a}=\vartheta_{cb}+\vartheta_{ba}$, changes $F_{ba}$ into $F_{ca}$ via a unitary transformation,
(16)$$F_{ca}=U_{cb}F_{ba},$$
with the matrix $U_{cb}$ given by
(17)$$U_{cb}=\left(\begin{array}{cc} \cos\vartheta_{cb} & \sin\vartheta_{cb}\\ -\sin\vartheta_{cb} & \cos\vartheta_{cb} \end{array}\right),$$
and $U_{cb}U_{cb}^{†}=1$. In terms of Pauli matrices, Equations (Equation 16) and (Equation 17) take the form
(18)$$F_{ba}=\cos\vartheta_{ba}\sigma_{z}-\sin\vartheta_{ba}\sigma_{x},$$
(19)$$U_{cb}=\cos\vartheta_{cb}I+i\sin\vartheta_{cb}\sigma_{y}.$$

Notice that when operating on $F_{ba}$, the matrix $U_{cb}$ leaves the right subindex *a* unchanged. This can be understood by noting that $U_{cb}$, being an orthogonal matrix, describes a rotation by an angle $\theta_{cb}$ around the a axis. Since
(20)$$U_{db}=U_{dc}U_{cb},$$
successive application of *U* on $F_{ba}$ gives
(21)$$U_{dc}U_{cb}F_{ba}=U_{dc}F_{ca}=U_{db}F_{ba}=F_{da}.$$

The same matrix *U*, when operating over a vector basis, transforms it into a new basis. Take, e.g., the initial basis of state vectors along b given by Equation (Equation 9) (with a→b), and apply to them the transformation $U_{cb}$,
(22)$$U_{cb}\left(\begin{array}{c} \cos\vartheta_{b}\\ -\sin\vartheta_{b} \end{array}\right)=\left(\begin{array}{c} \cos\vartheta_{c}\\ -\sin\vartheta_{c} \end{array}\right),\;U_{cb}\left(\begin{array}{c} \sin\vartheta_{b}\\ \cos\vartheta_{b} \end{array}\right)=\left(\begin{array}{c} \sin\vartheta_{c}\\ \cos\vartheta_{c} \end{array}\right).$$

Therefore, the change in the measuring context from (a,b) to (a,c) also implies a change in vector basis from ${|\pm \rangle }_{b}$ to ${|\pm \rangle }_{c}$.

Notice that this transformation does not have any effect on the state of the system. It does, however, introduce a change in the *partitioning* of the probability space, reflected in the coefficients of the probability matrix (Equation 12).

### 2.4. Context-Dependent Partitioning of the Probability Space

Given that the two alternative expressions for the correlation given by Equation (Equation 8) involve the probabilities conditioned by a sign of the projection along a, we may choose the first expression,
(23)$$C_{Q}\left(a,b\right)=P_{ab}\left(+|+\right)-P_{ab}\left(-|+\right),$$
which contains the probabilities of the results obtained for *b* conditioned by a=+1. In statistical terms, this means that instead of the entire ensemble represented by $|\psi \rangle$, only the subensemble of cases for which a=+1 is being considered; this subensemble is represented by the state vector ${|+\rangle }_{a}$.

We may therefore carry out an alternative calculation of $C_{Q}\left(a,b\right)$ by writing
(24)$$C_{Q}\left(a,b\right)={\langle +|}_{a}\left(\hat{\sigma }\cdot b\right)\left(\hat{\sigma }\cdot a\right){|+\rangle }_{a},$$
and inserting the spectral decomposition in terms of the projection operators
(25)$${\hat{P}}^{\pm }\left(b\right)={|\pm \rangle }_{b}{\langle \pm |}_{b},$$
with eigenvalues
(26)$$b^{\pm }=\pm 1,$$
so that Equation (Equation 24) gives, on account of Equations (10) and (11),
$$C_{Q}\left(a,b\right)={\langle +|}_{a}\left({|+\rangle }_{b}{\langle +|}_{b}-{|-\rangle }_{b}{\langle -|}_{b}\right){|+\rangle }_{a}$$
(27)$$=P_{ab}\left(+|+\right)-P_{ab}\left(-|+\right)=\cos^{2}\vartheta_{ba}-\sin^{2}\vartheta_{ba}.$$

Each of the terms within the parentheses in the upper line of this equation projects onto one of two mutually orthogonal and complementary subspaces $U^{\pm }\left(b=\pm |a=+\right)$ that add to form space *S* [11],
(28)$$S\left(b|+\right)=U^{+}\left(+|+\right)⊕U^{-}\left(-|+\right).$$

In operational terms ([12], Ch. 2), this means that the result of every measurement of the projection along b falls under one and only one of these (eigen) subspaces. Each of the terms $P_{ab}\left(\pm |+\right)$ represents a probability measure, namely the probability of obtaining $b^{\pm }$ as the result of a measurement, in accordance with the Born rule ([13], Ch. 1). This assigns an unambiguous meaning to the term *measurement dependence*: it refers to the dependence of the *partitioning of the probability space* on the measurement setting. This will become clear with the illustration presented in Section 4.

## 3. Probability Distribution for the Electron Spin

In a recent article [4], a general probability distribution ρ(ϕ) was proposed for the electron spin projection problem, which serves to reproduce the conditional probabilities and the quantum correlation $C_{Q}\left(a,b\right)$. This probability distribution has the form (the same formula for the distribution, Equation (Equation 29), has been previously obtained by other authors, also within the standard framework of quantum mechanics; see, e.g., [14])
(29)$$\rho \left(\phi \right)=\frac{1}{2}\sin\phi ,\;0\leq \phi \leq \pi ,$$
with
(30)$$\int_{\Phi }\rho \left(\phi \right)d\phi =1.$$

The discussion on the physical meaning of ϕ is left for the following section; for the time being, ϕ represents an independent variable, the value of which may vary from realization to realization, within the interval $\left[0,\pi \right]$. The partitioning of the probability space Φ into $\Phi_{ab}^{+}$, $\Phi_{ab}^{-}$ must be such that, according to Equation (11),
(31)$$\int_{\Phi_{ab}^{+}}\rho \left(\phi \right)d\phi =\cos^{2}\vartheta {}_{ba},\;\int_{\Phi_{ab}^{-}}\rho \left(\phi \right)d\phi =\sin^{2}\vartheta {}_{ba}.$$

With ρ(ϕ) given by Equation (Equation 29), the subdivision is (recall that $\vartheta_{ba}=\theta_{ba}/2$)
(32a)$$\int_{\Phi_{ab}^{+}}\rho \left(\phi \right)d\phi =\frac{1}{2}\int {{}_{\theta_{ba}}}^{\pi }\sin\phi d\phi =\cos^{2}\frac{\theta_{ba}}{2},$$
(32b)$$\int_{\Phi_{ab}^{-}}\rho \left(\phi \right)d\phi =\frac{1}{2}\int_{0}{}^{\theta_{ba}}\sin\phi d\phi =\sin^{2}\frac{\theta_{ba}}{2}.$$

The correlation is given accordingly by
(33)$$C_{Q}\left(a,b\right)=\left(\int_{\Phi_{ab}^{+}}-\int_{\Phi_{ab}^{-}}\right)\rho \left(\phi \right)d\phi =\cos\theta {}_{ba},$$

Equation (Equation 29) can therefore be considered to represent a bona fide ‘hidden-variable’ distribution.

It is important to keep in mind that the contextuality resides in the partitioning of the sample space. In other words, the same function ρ(ϕ) applies to different settings (b or $b^{\prime }$); but the *complementary intervals* of values of ϕ that give either +1 or −1 depend on the setting, and therefore correspond to *different realizations of the sample space*.

## 4. Geometric Model for the Electron Spin

The form of the probability distribution (Equation 29), along with the partitioning of the sample space indicated in Equations (32), is suggestive of a geometric representation that can be explored as a basis for a model for the spinning electron [4].

In line with the probabilistic description discussed above, we consider an ensemble of realizations. Assume we want to determine the sign of *b*, *given a certain value fora*, say a=+1. This means that all the elements of the ensemble considered, if actually measured (projected) along a, would give the result a=+1. We are assuming all relevant vectors to lie on the xz plane, for simplicity, taking into account that our probability density depends on one variable only. The direction a may be aligned with the +z axis, and b is then contained in the *xz* plane, forming an angle $\theta_{ba}$ with the *z* axis.

Having defined our ensemble as above, we know for sure that a spin projection along a will always give a=+1. In terms of the conditional probabilities introduced in Section 2, $P_{aa}\left(+|+\right)=1,\;P_{aa}\left(-|+\right)=0.$

This means that in all cases corresponding to this ensemble, the spin vector must lie in the upper (or northern) half plane, forming in principle any angle measured on the xz plane. *We propose to identify the variable ϕ with that angle*; then, ϕ lies in the interval 0≤ϕ≤π, with the origin of ϕ along the −x axis and ϕ increasing clockwise; see Figure 1. (Conversely, for the complementary ensemble defined by a=−1, one would have $P_{aa}\left(-|-\right)=1,\;P_{aa}\left(+|-\right)=0$, and for every realization the spin vector would lie somewhere in the lower half plane. The argument is of course reversible, in the sense that if *b* is given, the angle variable ϕ is measured with reference to the direction of b).

It is important to bear in mind the distinction between the spin direction, defined by ϕ, and its projections along a and b, which have well-defined signs ±1. An actual spin measurement, say along a, would of course affect the spin by projecting it along that axis, thereby preventing a second measurement (say along b) from being carried out on the original spin state. In other words, the quantum single-spin correlation (Equation 14) cannot be tested through this measurement procedure.

When a=+1, the sign of the projection along a direction b lying on the xz plane and forming an angle $\theta_{ba}$ with the +z axis is b=+1 for any angle ϕ on that plane such that $\theta_{ba}<\phi \leq \pi$, whilst it is negative for $0\leq \phi <\theta_{ba}$. This gives a concrete geometrical meaning to Equations (Equation 31)–(Equation 33) and justifies the partitioning of the probability space into the complementary subspaces $\Phi_{ab}^{+}\left(\theta_{ba},\pi \right)$, $\Phi_{ab}^{-}\left(0,\theta_{ba}\right)$.

What determines in each individual instance the specific value of the variable ϕ—i.e., the orientation of s—is not known here; ϕ may vary at random between realizations within the entire interval $\left[o,\pi \right].$ What the source of such randomness is and the mechanism that gives rise to the distribution function ρ(ϕ) is also unknown at this stage. What is important here is that a bona fide probability distribution exists that reproduces the desired conditional probabilities and correlations without additional assumptions.

To make the context dependence more explicit, one may rewrite Equation (Equation 33) as
$$C_{Q}\left(a,b\right)=\int_{\Phi }b\left(\phi \right)\rho \left(\phi \right)d\phi =\frac{1}{2}\left(\int {{}_{\theta_{ba}}}^{\pi }-\int_{0}{}^{\theta_{ba}}\right)\sin\phi d\phi$$
(34)$$=\frac{1}{2}\int_{0}^{\pi }\left[sign\sin(\phi -\theta {}_{ba})\right]\sin\phi d\phi =\cos\theta_{ab}.$$

When the direction is changed from b to $b^{\prime }$, the geometry changes, and the probability space is subdivided accordingly, so that one obtains instead
(35)$$C_{Q}\left(a,b^{\prime }\right)=\frac{1}{2}\int_{0}^{\pi }\left[sign\sin(\phi^{\prime }-\theta {}_{b^{\prime }a})\right]\sin\phi^{\prime }d\phi^{\prime }=\cos\theta {}_{b^{\prime }a}.$$

A prime has been added to the integration variable ϕ in Equation (Equation 35) to stress that, although the distribution *function* ρ(ϕ) is the same, its *realization* is independent from the previous one. This means that the individual results obtained in one context *may not be transferred to the other*.

The observation just made has important implications: it ascribes an unavoidable random character to the variable ϕ. If the behavior of the system were deterministic, one could label every individual element of the ensemble and assign to it a fixed value of ϕ, regardless of which projection (whether along b or $b^{\prime }$) is being measured.

## 5. The Entangled (Singlet) Bipartite System

### 5.1. Separating the Contributions to the Spin Correlation

We shall now apply the above reasoning to the bipartite system in an entagled singlet spin state. In preparation for this, we write the singlet state vector $|\Psi^{0}\rangle$ in terms of the standard notation $|\phi \rangle |\chi \rangle =|\phi \rangle ⊗|\chi \rangle ,$ with $|\phi \rangle$ a vector in the Hilbert space of spin 1, and $|\chi \rangle$ a vector in the Hilbert space of spin 2,
(36)$$|\Psi^{0}\rangle =\frac{1}{\sqrt{2}}\left(|{+}_{r}\rangle |{-}_{r}\rangle -|{-}_{r}\rangle |{+}_{r}\rangle \right),$$
where the direction r is arbitrary since the singlet state is spherically symmetric. The projection of the spin 1 operator along a is described by $(\hat{\sigma }\cdot a)⊗I$, and the projection of the spin 2 operator along b is described by $I⊗(\hat{\sigma }\cdot b)$, whence the correlation is given by
(37)$$C_{Q}\left(a,b\right)=\langle \Psi^{0}|\left(\hat{\sigma }\cdot a\right)⊗\left(\hat{\sigma }\cdot b\right)|\Psi^{0}\rangle .$$

For the calculation of $C_{Q}$, we make use of the individual spin state vectors (Equation 9) to construct an orthonormal basis for the bipartite system [3]:
(38)$${|\phi^{1}\rangle }_{ab}={|+\rangle }_{a}{|-\rangle }_{b},\;\;{|\phi^{2}\rangle }_{ab}={|-\rangle }_{a}{|+\rangle }_{b},\;{|\phi^{3}\rangle }_{ab}={|+\rangle }_{a}{|+\rangle }_{b},\;\;{|\phi^{4}\rangle }_{ab}={|-\rangle }_{a}{|-\rangle }_{b},$$
and write
(39)$$C_{Q}\left(a,b\right)=\langle \Psi^{0}|\left(\hat{\sigma }\cdot a\right)\left(\sum_{k=1}^{4}{|\phi^{k}\rangle }_{ab}{\langle \phi^{k}|}_{ab}\right)\left(\hat{\sigma }\cdot b\right)|\Psi^{0}\rangle .$$

The operators
(40)$${\hat{P}}^{k}\left(a,b\right)={|\phi^{k}\rangle }_{ab}{\langle \phi^{k}|}_{ab}$$
appearing in (Equation 39) are the projection operators in the product space of the individual spin spaces, ${S=S}_{1}⊗S_{2}$, with respective eigenvalues $A_{k}$ corresponding to the bipartite states ${|\phi^{k}\rangle }_{ab}$ and given according to (Equation 38) by
(41)$$A_{1}=A_{2}=-1\equiv A^{-},\;A_{3}=A_{4}=+1\equiv A^{+}.$$

This allows us to rewrite Equation (Equation 39) in the form
(42)$$C_{Q}\left(a,b\right)=\sum_{k=1}^{4}A_{k}\left(a,b\right)C_{k}\left(a,b\right),$$
which is the appropriate spectral decomposition of the spin correlation. In terms of the projection operators (Equation 40), we may write the spin correlation operator in the form
(43)$${\hat{C}}_{Q}\left(a,b\right)=\sum_{k=1}^{4}A_{k}\left(a,b\right){\hat{P}}^{k}\left(a,b\right),$$
with $A_{k}$ being the eigenvalues given by Equations (Equation 41). The coefficients appearing in (Equation 42)
(44)$$C_{k}\left(a,b\right)=|\left(\langle \phi^{k}{|}_{ab}\right)|\Psi^{0}{\rangle |}^{2},$$
representing the relative weights of the eigenvalues $A_{k}$, are calculated with the help of Equations (Equation 37) and (Equation 38),
(45a)$$C_{1}\left(a,b\right)=C_{2}\left(a,b\right)=\frac{1}{2}\cos^{2}\vartheta_{ba},$$
(45b)$$C_{3}\left(a,b\right)=C_{4}\left(a,b\right)=\frac{1}{2}\sin^{2}\vartheta_{ba}.$$

The conditional probabilities are therefore given in this case by
(46a)$$P_{ab}\left(+|-\right)=P_{ab}\left(-|+\right)=\cos^{2}\vartheta_{ba},$$
(46b)$$P_{ab}\left(+|+\right)=P_{ab}\left(-|-\right)=\sin^{2}\vartheta_{ba},$$
whence the matrix of probabilities (Equation 1) becomes
(47)$$P\left(b,a\right)=\left(\begin{array}{cc} \sin^{2}\vartheta_{ba} & \cos^{2}\vartheta_{ba}\\ \cos^{2}\vartheta_{ba} & \sin^{2}\vartheta_{ba} \end{array}\right).$$

Equations (45) inserted into (Equation 42) reproduce the quantum result,
(48)$$C_{Q}\left(a,b\right)=-\cos\theta_{ba}.$$

### 5.2. Context-Dependent Partitioning of the Probability Space in the Bipartite Case

We observe that each term in the sum (Equation 43) projects onto one and only one of the four mutually orthogonal subspaces $U^{k}\left(a,b\right)$ that add to form space S [11],
(49)$$S=U^{1}⊕U^{2}⊕U^{3}⊕U^{4}.$$

In operational terms ([12], Ch. 2), this means that the result of every (joint) measurement falls under one and only one of these (eigen)subspaces. Each of the coefficients $C_{k}$ is therefore identified with a probability measure, namely the probability of obtaining $A_{k}$ as the result of a measurement.

For the observable $C_{Q}\left(a,b^{\prime }\right)$ with $b^{\prime }\neq b$, the projection operators are
(50)$${\hat{P}}^{k}\left(a,b^{\prime }\right)={|\phi^{k}\rangle }_{ab^{\prime }}{\langle \phi^{k}|}_{ab^{\prime }},$$
where ${|\phi^{k}\rangle }_{ab^{\prime }}$ is defined as in (Equation 38), with *b* replaced by $b^{\prime }$. Therefore, instead of (Equation 49), the spectral decomposition involves now the partitioning into four orthogonal subspaces $U^{k}\left(a,b^{\prime }\right)$, such that every (joint) measurement falls under one and only one of these new subspaces.

Again, we note that the probability subspaces are specific to the measurement setting, thus endowing the term *measurement dependence* used in the context of the Bell-type inequalities (see, e.g., [15]) with a clear meaning: it refers to the dependence of the *partitioning of the probability space* on the setting.

We now proceed to carry out an alternative calculation of $C_{Q}\left(a,b\right)$, taking advantage of the fact that the elements in (Equation 47) are conditional probabilities, with the rows containing the probabilities for b=±1 given a=±1, and the columns containing those for a=±1 given b=±1 (or vice versa). On account of (Equation 8), we may write the correlation in terms of the conditional probabilities b=±1 given a=1,
(51)$$C_{Q}\left(a,b\right)=P_{ab}\left(+|+\right)-P_{ab}\left(-|+\right)=\sin^{2}\vartheta_{ba}-\cos^{2}\vartheta_{ba}=-\cos\theta_{ba}.$$

Expressed in statistical terms, instead of the entire ensemble represented by $|\psi \rangle$, only the subensemble of cases for which a=+1 is being considered; this subensemble is represented by the state vector ${|+\rangle }_{a}$. This implies restricting the sum in (Equation 39) to those terms in (Equation 38) that involve ${|+\rangle }_{a}$, i.e.,
$$C_{Q}\left(a,b\right)=\langle \Psi^{0}|\left(\hat{\sigma }\cdot a\right)\left({\hat{P}}^{1}+{\hat{P}}^{3}\right)\left(\hat{\sigma }\cdot b\right)|\Psi^{0}\rangle ,$$
with $A_{1}=-1$ and $A_{3}=+1$, according to (Equation 41). We are working now in the subspace formed by
$$S^{a+}=U^{1}⊕U^{3},$$
which, in operational terms, means that the result of every measurement of *b* falls under either $U^{1}$ or $U^{3}$. The partitioning must be such that
(52)$$\int_{\Phi_{ab}^{-}}\rho \left(\phi \right)d\phi =\cos^{2}\vartheta {}_{ba},\;\int_{\Phi_{ab}^{+}}\rho \left(\phi \right)d\phi =\sin^{2}\vartheta {}_{ba},$$
which gives, using Equation (Equation 29) for the probability distribution,
(53a)$$\int_{\Phi_{ab}^{-}}\rho \left(\phi \right)d\phi =\frac{1}{2}\int {{}_{\theta_{ba}}}^{\pi }\sin\phi d\phi ,$$
(53b)$$\int_{\Phi_{ab}^{+}}\rho \left(\phi \right)d\phi =\frac{1}{2}\int_{0}{}^{\theta_{ba}}\sin\phi d\phi ,$$
whence $C_{Q}\left(a,b\right)=-\cos\theta_{ba}.$

### 5.3. Application of the Geometric Model to the Bipartite Case

A procedure analogous to the one applied in the single-spin case in Section 4 can be followed, mutatis mutandis, in the entangled bipartite case. This is made possible thanks to the dependence of the correlation on the *conditioned probabilities*, as expressed in the general Formula (Equation 8). In the present case, the spin projections along a and b are those of particles 1 and 2, and the probabilities are conditioned by the sign of the projection of spin 1, spin 2 being *antiparallel* to spin 1. Since we have chosen a=+1, we may refer for illustration to the geometry used in Section 4 and simply invert the sign of b, to reflect the fact that the spins are now antiparallel. The corresponding change of sign of *b* is reflected in the final outcome, $C\left(a,b\right)=-\cos\theta_{ba}$ (Equation (Equation 34)).

As discussed at the end of Section 4 in connection with the single-spin case, it is important to bear in mind the distinction between a spin direction and its projection, which has a well-defined sign ±1. However, contrary to the single-spin case, when dealing with a couple of entangled spins, an actual measurement of spin 1 that projects it along a does not affect spin 2, which is still in its original orientation and can therefore be measured (projected) simultaneously (i.e., in the same run). This is the basis for the numerous measurements that have been carried out to test the quantum bipartite spin correlation (Equation 48).

In the conventional terminology, the conclusion is that the ‘hidden’ variable ϕ with its associated distribution ρ(ϕ) does not serve to complete the quantum description, since the random element is still present. It does serve, however, to reproduce the correct quantum correlation formulas, thereby offering a better understanding of the probabilistic features of spin within the context of standard probability theory and providing a geometric explanation for such features.

To demonstrate that there is indeed no need to abandon classical probability has also been the motivation behind different computer simulations that produce results in violation of Bell-type inequalities (e.g., [16,17,18,19]; see also [20] and further references therein). As indicated in Ref. [17], ’one should not try to explain away the strange features of quantum mechanics as some kind of defect of classical probabilistic thinking, but one should use classical probabilistic thinking to pinpoint these features’. The present work offers a contribution in this direction.

## 6. Final Comment: A Possible Physical Picture of Spin

At this point one may ask whether a physical image of the electron spin can be made compatible with the geometric representation just discussed, under the condition that ρ(ϕ), with 0≤ϕ≤π, represents a distribution of random variables associated with the individual realizations of the spin orientation within the ensemble. Such an image would have to be consistent with the physical notion of spin as a dynamical quantity, with an associated intrinsic angular momentum s of fixed magnitude and a magnetic moment roughly given by μ=(e/m)s.

In the presence of a constant magnetic field H, a classical, frictionless magnetic spinning body is known to regularly precess around the direction of H with constant angular frequency as a result of the torque exerted by the field (see, e.g., Ref. [21]). A similar image has been conventionally associated with the electron, in which case the frequency of precession or Larmor frequency is given by $\omega_{L}=\left(e/m\right)H$. Even for intense magnetic fields, this frequency is many orders of magnitude smaller than the spinning frequency associated with the zitterbeweung predicted by Dirac’s equation, which is estimated to be of the order of Compton’s frequency, $\omega_{C}=mc^{2}/\hbar$∼$10^{21}$ s${}^{-1}$ (see, e.g., [22,23,24], and further references therein).

This crude image does not seem to leave any space for the additional inclination variable represented by ϕ in our geometric model, and even less for the possible random character of this variable. However, such a picture may change in the light of recent experimental evidence. Observations made with ferromagnetic materials in the pico- and femtosecond scales ([25]; see also [26]), provide evidence of a spin dynamics far richer than previously assumed due to effects of damping and inertia. This also makes the study of the dynamics markedly more complicated owing to the nonlinearity of the dynamical equations, which are impossible to solve analytically. The analysis of the detailed dynamics of the spinning electron is clearly outside the scope of the present paper. However, of relevance to our discussion is the theoretical possibility of spin nutations, similar to the ones of a spinning top, and their experimentally observed appearance at a characteristic frequency $\omega_{N}$ much higher than the usual Larmor precession, yet much smaller than Compton’s frequency. These apparently intrinsic nutations have been established experimentally thanks to the use of an intense, transient magnetic field from a superradiant source of frequency close to 10${}^{12}$ s, to which the nutating spin resonates. The lack of such sources had previously hampered the observation of this nutation dynamic.

Take now the geometric model described in the previous section and consider the dynamics of the electron spinning around its own axis plus the spin angular momentum precessing around the direction of the magnetic field along the *z* axis, which was defined as the direction a. If, in addition, the spin vector is allowed to nutate, and it does so in a highly complex and irregular manner due to the nonlinearity of the dynamics, it may in principle scan the entire range of values of ϕ, from 0 to π. As long as we cannot observe this nutation, because of its extremely high frequency, the angle ϕ remains as a ’hidden variable’. We are not able to determine the variations of ϕ that occur with such high resolution; we only know that, on average, they must be described by a distribution function such as ρ(ϕ). Whether the randomness of ϕ is due to the permanent interaction of the spinning electron with the fluctuating vacuum or whether it is a product of the chaotic behavior of spin at this scale is an open question; in any case, there is no need to think that randomness is an inherent element of physics.

With this discussion, we hope to have provided elements in favor of the plausibility of a physical explanation for the probabilistic description of the electron spin given by quantum mechanics, thereby avoiding the need to resort to arguments of an unphysical or spooky nature. To conclude, we may briefly say that, although the electron spin itself is a quantum property whose dynamics is still in need of a more complete theory, the current probability theory seems well suited for an explanation of its probabilistic features. 

## Figures and Tables

**Figure 1 entropy-24-01439-f001:**
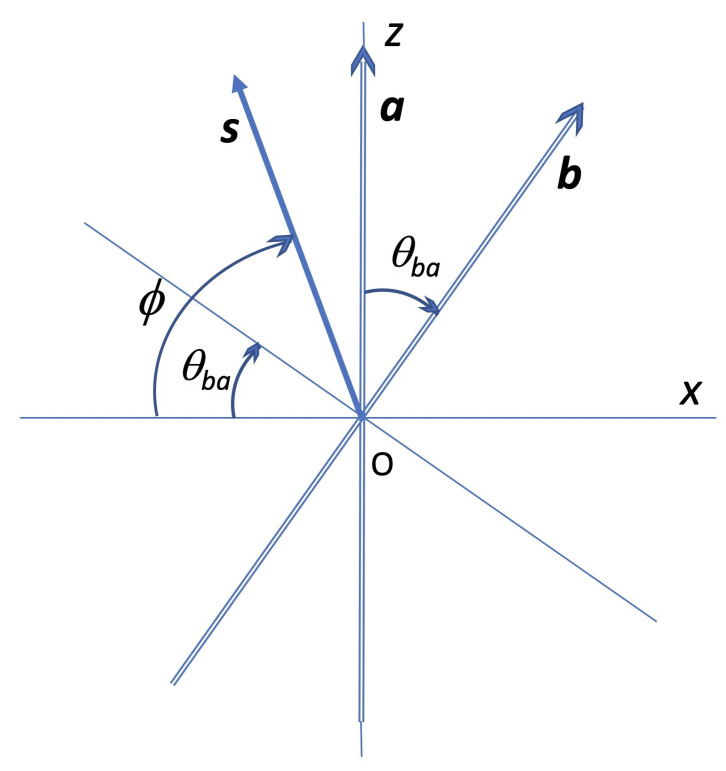
The vectors a,b define the measurement setting. For any vector s lying in the upper half plane (i.e., 0≤ϕ≤π), the sign of its projection along a is positive. The sign of its projection along b is positive for $\theta_{ba}<\phi \leq \pi$, and negative for $0\leq \phi <\theta_{ba}$.

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
