# Peer review of "Electron Spin Correlations: Probabilistic Description and Geometric Representation"

_entropy, 2022, doi:10.3390/e24101439_

Round 1

Reviewer 1 Report

I found this paper to be interesting and well written. The author does a good job of clarifying the concepts of contextuality and measurement dependence. I found only one typo. On page 2 line -10, P(a.b) should be P(a,b). I recommend that the paper be published.

Author Response

I thank the reviewer for his positive comments.

The typo noted by the reviewer has been corrected.

Reviewer 2 Report

The manuscript develops the probabilistic interpretation of quantum mechanics using one- and two-qubit models. I believe, it can be of interest for some readers and, thus, can be published, but, unfortunately, I cannot understand what is the main result of the manuscript. I think, it would be better if the author state more clearly what is the main contribution of this paper.

Other major comments:

1. Eq. (15). What is the physical meaning of F? The physical meaning of U is the transformation rule for F, but what is F? Moreover, it seems that F is define ambiguously because of the choice of the signs +/-. Please, comment this.

2. The beginning of Sec. 3. What is the physical meaning of phi?

3. Sec. 4. Unfortunately, I cannot understand the geometric model.

(a) At first, a figure is required here.

(b) Also, what do we draw on the sphere? Do we draw a state vector? Then what is the difference with the Bloch sphere? 

(c) If we align the direction a with the +z axis and a=+1, then why the spin vector is merely in the northern hemisphere? It seems that it should be exactly in the north pole. A picture as well as more clear explanations are strictly required.

Author Response

I thank the reviewer for his comments and suggestions, which have helped me state more clearly the purpose of the paper and its main contributions, as well as reinforce important arguments.

In response to the reviewer’s introductory comment, I have completed and reordered the text of the abstract and made small changes in the introduction.

Response to specific comments:

  • The matrix F was introduced in section 2.3 as a mathematical tool to certify the correct properties of the unitary transformation as required by the consistency argument in Ref. (5). This is now explicitly said in the paragraph preceding Eq. (15).
  • In section 3, phi simply represents the independent variable on which the probability distribution depends, according to Eq. (29); the purpose of section 4 is precisely to associate a meaning to it. A sentence to this effect has been added following Eq. (30).
  • a) Figure 1, as well as some additional explanations, have been included in section 4 to facilitate understanding of the discussion of the geometric model and the meaning of phi.
  1. b) As shown in figure 1, all relevant vectors, including s, lie in the xz plane, the angle phi being the single variable considered. I have corrected the terminology by replacing ‘hemisphere’ with ‘half plane’.
  2. c) More explicit and detailed explanations have been introduced in various parts of sections 4 and 5 to stress the difference between the spin vector and its projections (or measurement outcomes). The paragraph preceding Figure 1 in section 3, as well as the second paragraph in section 5.3, have been added precisely to elaborate on this important distinction and its implications.

Round 2

Reviewer 2 Report

The authors have addressed all my comments. The paper can be published in the present form.